# FreeFlow: Latent Flow Matching for Free Energy Difference Estimation

## Abstract

Estimating free energy differences between molecular systems is fundamental for understanding molecular interactions and accelerating drug discovery. Current techniques use molecular dynamics to sample the Boltzmann distributions of the two systems and of several intermediate "alchemical" distributions that interpolate between them. From the resulting ensembles, free energy differences can be estimated by averaging importance weight analogs for multiple distributions. Instead of time-intensive simulations of intermediate alchemical systems, we learn a fast-to-train flow to bridge the two systems of interest. After training, we obtain free energy differences by integrating the flow's instantaneous change of variables when transporting samples between the two distributions. To map between molecular systems with different numbers of atoms, we replace the previous solutions of simulating auxiliary "dummy atoms" by additionally training two autoencoders that project the systems to a same-dimensional latent space in which our flow operates. A generalized change of variables formula for trans-dimensional mappings allows us to employ the dimensionality collapsing and expanding autoencoders in our free energy estimation pipeline. We validate our approach on systems of increasing complexity: mapping between Gaussians, between subspaces of alanine dipeptide, and between pharmaceutically relevant ligands in solvent. All results show strong agreement with reference values. We provide an example anonymized Jupyter notebook for our method applied to Gaussian distributions here.

## 1 Introduction

Estimating free energy differences between two states of a thermodynamic system allows us to compare the relative likelihoods of the two states (Chipot et al., 2007; Stoltz et al., 2010). This task underpins insights in computational chemistry, biology, and is extensively used in drug discovery, where free energy differences can inform which ligand is more likely to bind to a protein. In this paper, we explore estimating free energy differences via a neural mapping that is based on flow matching (Lipman et al., 2023; Albergo et al., 2023; Liu et al., 2022) between the Boltzmann distributions of the two molecular systems of interest.

In the free energy difference estimation problem (see Figure 1), we are given two molecular systems, A and B, and their unnormalized densities (energy functions) over their 3D structures. Their free energy difference is $\Delta F = -k_B T \ln(Z_B/Z_A)$ where $Z_A$ and $Z_B$ are their normalizing constants. For instance, in the context of drug discovery, the systems A and B could be two different molecules bound to the same protein. Their free energy difference (together with the molecules' free energy differences in solvent) determines the difference in

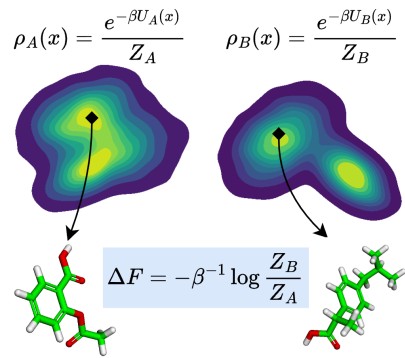

$$\rho_A(x) = \frac{e^{-\beta U_A(x)}}{Z_A} \qquad \rho_B(x) = \frac{e^{-\beta U_B(x)}}{Z_B}$$

$$\Delta F = -\beta^{-1} \log \frac{Z_B}{Z_A}$$

Figure 1: Free energy difference as the log ratio of two Boltzmann distributions' normalizing constants. Intractability of the normalizing constants makes the problem challenging.

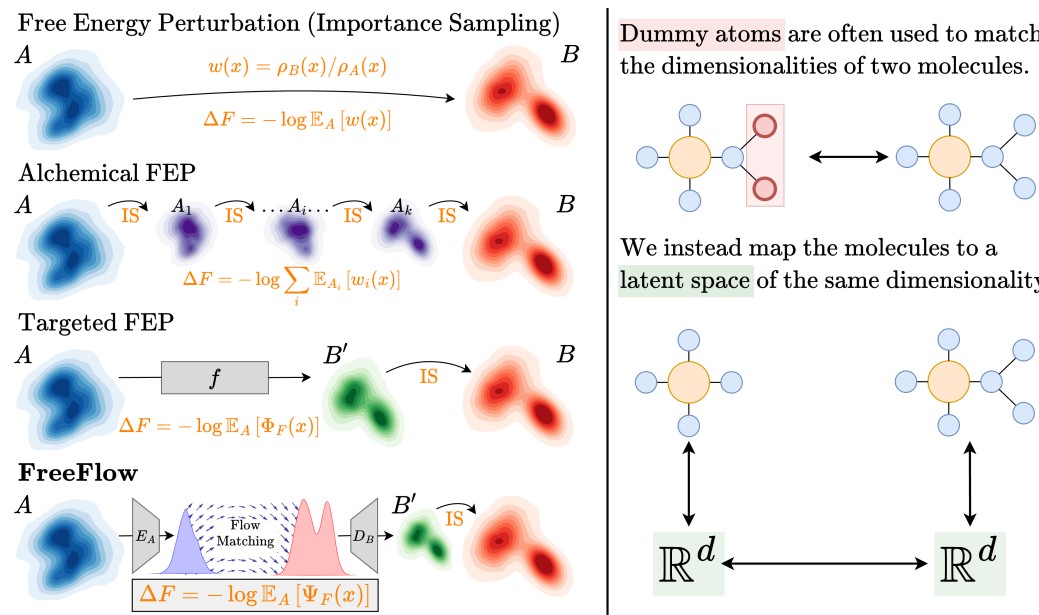

Figure 2: **Overview of approaches for estimating free energy differences between systems** $A$ **and** $B$. *Free Energy Perturbation*: importance sampling between the two systems with samples obtained via molecular dynamics simulations, and the negative log of the average importance weight estimates the free energy difference. *Alchemical FEP*: importance sampling between a sequence of systems interpolating from $A$ to $B$. Increased overlap between the subsequent systems makes importance sampling converge faster at the cost of running separate simulations for each intermediate system. *Targeted FEP*: a learned mapping is used instead to map $A$ to $B'$, which is expected to have a higher overlap with $B$. *FreeFlow*: encode $A$ and $B$ into a low dimensional latent space and learn a latent mapping to be able to bridge systems of different dimensions. On the right we highlight that traditional methods often materialize non-physical systems by the addition of "dummy atoms" to be able to bridge between systems of different dimensions, which FreeFlow avoids by mapping both systems to a fixed-dimensional latent space.

their binding affinities to the protein. Thus, access to estimates between a set of molecules allows for identifying the strongest binder out of a pool of candidate molecules.

The common traditional approaches for such estimations are based on Free Energy Perturbation (FEP). The FEP identity (Zwanzig, 1954) reduces estimating free energy differences to importance sampling between distributions A and B: the negative log of the average importance weights is the free energy difference. For molecules, the distributions are sampled by running molecular dynamics (MD) simulations. The convergence of this estimate depends on the variance of the importance weights, which is large if there is insufficient overlap between distributions A and B. Thus, in practice, one turns to *alchemical FEP* (Bash et al., 1987; Mey et al., 2020) where a series (typically a few tens) of "alchemical" molecular systems between systems A and B are simulated (see Figure 2). Modeling these additional intermediate distributions yields a lower variance free energy difference estimate at the cost of additional molecular dynamics simulation time.

Instead of bridging distributions A and B via additional MD simulations, FreeFlow learns a neural map between them and estimates their free energy differences by observing the change of density when transporting samples from system A to system B or back. The aim is to *overfit* and run inference with such a map faster than carrying out a series of MD simulations for FEP (while maintaining or improving upon the accuracy of FEP). For this purpose, previous work (Wirnsberger et al., 2020) employed normalizing flows (Rezende & Mohamed, 2016) with the same input and output dimensionality that do not accommodate different dimensionality (different numbers of atoms) between systems A and B.

Concretely, we propose FreeFlow to map between distributions of arbitrary dimensions by encoding the systems into a lower dimensional latent space and learning a flow model in that latent space using flow matching. The autoencoders to produce the latent spaces for systems A and B are fast-to-train and overfitting a flow between their small dimensional latent spaces is equally efficient. After training this map, computing the free energy difference requires evaluating the change of density a sample incurs when transporting it between the distributions. While this is trivial for normalizing flows, FreeFlow involves changes in dimensionality, which we accommodate with a *generalized change of variables formula* for "trans-dimensional" mappings.

Empirically, we evaluate FreeFlow on a series of free energy difference estimations of increasing complexity. First, we confirm that FreeFlow is able to recover analytically computed free energy differences between Gaussians of *different dimensionalities*. Next, we turn to a well-explored molecular system, alanine dipeptide, and estimate free energy differences between partitions of its state space. Lastly, we tackle the real-world task of computing free energy differences between different pharmaceutically relevant ligands in solvent. In this experiment, we observe Spearman and Pearson correlations of up to 0.93 between our free energy difference and the Free Energy Perturbation reference values.

We summarize our key contributions as:

1. A simulation-free continuous normalizing flow training procedure based on flow matching without constraints such as easy-to-compute Jacobian determinant or fast invertibility unlike older flows.

2. A map that translates between systems of arbitrary dimensionality via a same-dimensional latent space, avoiding the introduction of dummy atoms and requiring an order of magnitude fewer MD simulations compared to intermediate-window-based methods such as Alchemical FEP.

3. Using a generalized change of variables formulation for computing density changes in trans-dimensional maps.

4. Validation on real-world pharmaceutically relevant ligands of varying numbers of atoms.

## 2 BACKGROUND

**Flow Matching.** Flow Matching (FM) (Lipman et al., 2023; Albergo et al., 2023; Liu et al., 2022) is a training framework for CNFs that avoids the need for simulation during training. Instead of integrating the ODE, FM directly trains the vector field $v_\theta(t, x)$ to match a target probability flow defined by a prescribed time-dependent probability path $p_t(x)$. The objective minimizes the discrepancy between the model's vector field and the target vector field that transports $p_t(x)$ along the flow:

$$\mathcal{L}_{\text{FM}}(\theta) = \mathbb{E}_{t, x \sim p_t} |u_t(x) - v_\theta(t, x)|^2, \tag{1}$$

where $u_t(x)$ is the target vector field derived from the continuity equation. To construct more expressive probability paths $p_t(x)$, Conditional Flow Matching (CFM) (Lipman et al., 2023; Tong et al., 2023) introduces a conditioning variable $z$ and expresses $p_t(x)$ as a combination of simpler distributions $p_t(x|z)$ such as Gaussians conditioned on $z$.

**Free Energy Calculations in Molecular Systems.** Free energy calculations are essential for understanding molecular interactions and predicting binding affinities in drug discovery (Chipot et al., 2007). The accuracy of these methods depends on the overlap between the configurations sampled from the different states. Insufficient overlap can lead to high variance and unreliable estimates. Techniques like stratification and the use of intermediate states help mitigate this issue but increase computational complexity.

The *Free Energy Perturbation* (FEP) method, introduced by Zwanzig (1954), provides an exact relationship for computing $\Delta F$ between two thermodynamic states $A$ and $B$:

$$\mathbb{E}_A \left[ e^{-\beta \Delta U(\mathbf{x})} \right] = e^{-\beta \Delta F}, \tag{2}$$

where $\beta = 1/(k_B T)$ is the inverse temperature, $\Delta U(\mathbf{x}) = U_B(\mathbf{x}) - U_A(\mathbf{x})$ is the potential energy difference at configuration $\mathbf{x}$, and $\mathbb{E}_A[\cdot]$ denotes the expectation over the equilibrium distribution

$\rho_A(\mathbf{x}) \propto e^{-\beta U_A(\mathbf{x})}$. However, the convergence of the FEP estimator critically depends on the overlap between the configurations sampled from $\rho_A$ and those relevant under $\rho_B$. Insufficient overlap can lead to high variance and slow convergence (Jarzynski, 2006).

To address this challenge, Jarzynski (2002) introduced *Targeted Free Energy Perturbation* (TFEP), which employs an invertible mapping $M : \mathcal{X} \rightarrow \mathcal{X}$ to transform configurations from $A$ to a new distribution $B'$, thereby increasing the overlap with $B$. The generalized FEP identity in TFEP is given by:

$$\mathbb{E}_A \left[ e^{-\beta \Phi_F(\mathbf{x})} \right] = e^{-\beta \Delta F}, \tag{3}$$

where the generalized energy difference $\Phi_F(\mathbf{x})$ is defined as:

$$\Phi_F(\mathbf{x}) = U_B(M(\mathbf{x})) - U_A(\mathbf{x}) - \beta^{-1} \log |\det J_M(\mathbf{x})|, \tag{4}$$

and $J_M(\mathbf{x})$ is the Jacobian matrix of $M$ at $\mathbf{x}$. By appropriately choosing $M$, one can enhance the overlap between $B'$ and $B$, improving the efficiency of the free energy estimation.

Building upon TFEP, Wirnsberger et al. (2020) introduced *Learned Free Energy Perturbation* (LFEP), using normalizing flows to learn the mapping $M$. Instead of relying on hand-crafted transformations, LFEP learns $M$ by optimizing a neural network to maximize the overlap between the transformed distribution $B'$ and the target distribution $B$. This approach provides a data-driven way to enhance free energy estimations without the need for explicit intermediate states or extensive physical intuition.

## 3 METHOD

Our aim is to estimate the free energy difference $\Delta F$ between thermodynamic systems $A$ and $B$ with equilibrium distributions $\rho_A$ and $\rho_B$, and potentially different numbers of atoms $n_A$ and $n_B$. For this purpose, we assume access to samples $\mathbf{x}_0 \sim \rho_A$ and $\mathbf{x}_1 \sim \rho_B$, which are obtained via molecular dynamics simulations when considering molecular systems. Unlike FEP, we do not carry out additional MD for intermediate alchemical systems to bridge between systems $A$ and $B$. Instead, we *overfit* a fast-to-train neural mapping to transport samples between them over a lower-dimensional latent space. Given this map, we estimate the free energy difference by transporting samples between $A$ and $B$ and averaging their incurred change of density.

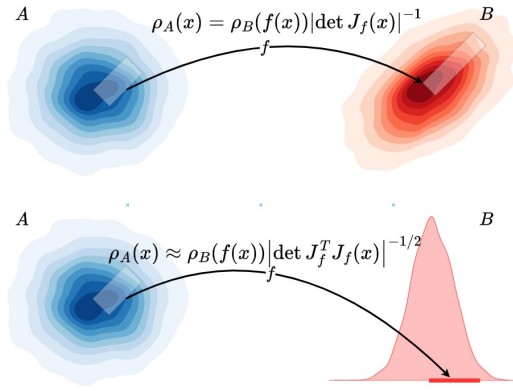

Concretely, our neural mapping consists of separate autoencoders $(E_A \circ D_A)$ and $(E_B \circ D_B)$ to map the samples from the two systems to the latent space and back, and an ODE parametrized via a flow model $v(x, t)$ between the latent spaces. Then we combine the encoder $E_A$, the

Figure 3: **The change of dimensionality problem.** The Jacobian of a trans-dimensional mapping is rectangular and hence does not have a determinant.

flow, and the decoder $D_B$ to map system $A$ to system $B$. In Section 3.1, we first summarize how neural maps (including ODEs parameterized as flow models) and their change of density can be employed to estimate free energy differences. Next, we present our autoencoders to map systems of arbitrary sizes $n_A, n_B$ to a fixed size same-dimensional latent space and how the change of density for such "trans-dimensional" maps (Section 3.2). Finally, in Section 3.3, we lay out our full free energy difference estimation procedure of 1) training autoencoders, 2) training a flow between their latent spaces, 3) and mapping samples from system A to system B while observing the map's change of variables.

### 3.1 Free Energy Differences via Neural Maps

We seek a mapping $f$ such that the pushforward distribution of $\rho_A$ through $f$ approximates $\rho_B$. Traditional normalizing flows (Rezende & Mohamed, 2016) model $f$ as a composition of invertible mappings with the likelihoods computed via the change of variables (CoV) formula as

$$\rho_B(\mathbf{x}_1) = \rho_A(\mathbf{x}_0) \left| \det\left( \frac{\partial f}{\partial \mathbf{x}} \right) \right|^{-1} \tag{5}$$

where $\mathbf{x}_1 = f(\mathbf{x}_0)$ and $\frac{\partial f}{\partial \mathbf{x}}$ is the Jacobian of $f$ at $\mathbf{x}_0$. We denote such models *discrete* normalizing flows. For discrete NFs, free energy differences can then be estimated via TFEP by computing the expectation in Equation 3 with

$$\Phi_F(\mathbf{x}) = U_B(f(\mathbf{x})) - U_A(\mathbf{x}) - \beta^{-1} \log \left| \det\left( \frac{\partial f}{\partial \mathbf{x}} \right) \right|. \tag{6}$$

However, normalizing flows requiring invertible components with efficiently-computable Jacobians might limit their expressivity. Flow matching on the other can be used with arbitrary neural networks as the flow model, and learn to map arbitrary distributions in simulation-free manner. It is thus an expressive yet efficient alternative to discrete normalizing flows.

Using flow matching, we train our normalizing flow between the same-dimensional latent representations of the two systems learned by our autoencoders, which are low-dimensional and hence lead to fast training, minimizing the objective

$$\mathcal{L}_{\mathrm{CFM}}(\theta) = \mathbb{E}_{t \sim \mathcal{U}(0,1),(\mathbf{x}_0,\mathbf{x}_1) \sim \pi(X_0,X_1),\mathbf{z}_t \sim p_t(E_A(x_0),E_B(x_1))} \left\| v_\theta(\mathbf{z}_t,t) - (E_B(\mathbf{x}_1) - E_A(\mathbf{x}_0)) \right\|_2^2 \tag{7}$$

where $\pi$ denotes the optimal transport coupling between the datasets $X_0, X_1$ approximated with mini-batches. Using OT couplings is advantageous as the learned vector field has straighter trajectories which lead to lower integration error. In particular for free energy estimation, approximating the OT map between $\rho_A$ and $\rho_B$ has been shown to result in paths with lower free energy, improving the convergence of the $\Delta F$ estimate (Decherchi & Cavalli, 2023).

The flow model leads to an ordinary differential equation (ODE) which we can integrate through time to transport the samples. For an ODE, the change in log-density w.r.t. time is given by the *instantaneous change of variables* formula (Chen et al., 2018)

$$\frac{\partial \log \rho(\mathbf{x}_t)}{\partial t} = - \operatorname{tr}\left( \frac{\partial v(\mathbf{x}_t,t)}{\partial \mathbf{x}_t} \right). \tag{8}$$

We integrate over time to obtain

$$\log \rho(\mathbf{x}_1) = \log \rho(\mathbf{x}_0) - \int_0^1 \operatorname{tr}\left( \frac{\partial v(\mathbf{x}_t,t)}{\partial \mathbf{x}_t} \right) dt \tag{9}$$

which we use to obtain free energy difference estimates by employing the following generalized energy difference to take the expectation over in Equation 3:

$$\Phi_F(\mathbf{x}) = U_B(\mathbf{x}_1) - U_A(\mathbf{x}_0) - \beta^{-1} \int_0^1 \operatorname{tr}\left( \frac{\partial v(\mathbf{x}_t,t)}{\partial \mathbf{x}_t} \right) dt. \tag{10}$$

### 3.2 Autoencoder Change of Variables

As noted above, we train our flow over a low-dimensional latent space, which enables fast training and allows FreeFlow to map between systems with different numbers of atoms. This is opposed to previous classical and ML solutions for estimating energy differences between systems of different dimensionality, which commonly simulate additional dummy-atoms in the lower-dimensional system.

Concretely, we first train two separate autoencoders, consisting of the encoders $E_A, E_B$ and the decoders $D_A, D_B$ for the two states $A$ and $B$. As the autoencoders are not required to generalize, we choose simple MLPs that map the flattened vectors of atom coordinates (ignoring the atom types) to the latent space $\mathcal{Z}$. In our experiments, we set this latent space to have 32 dimensions. We train

our autoencoders until the reconstruction MSE converges on training data since we will be using the training data to estimate $\Delta F$. We chose MLPs instead of more popular architectures for molecular representation learning such as equivariant graph neural networks after validating their performance on alanine dipeptide (see Figure 7). The MLP achieved a lower reconstruction error, while being 8x faster in terms of training speed.

Given these autoencoders, the end-to-end mapping from $A$ to $B$ can then be expressed as

$$\mathbf{x}_1 = f(\mathbf{x}_0) = D_B\left(\text{ODE}\left(E_A(\mathbf{x}_0)\right)\right) \tag{11}$$

where ODE denotes integrating $v_\theta(\mathbf{z}_t, t)$ starting from $\mathbf{z}_0 = E_A(\mathbf{x}_0)$, i.e., $\mathbf{z}_0 + \int_0^1 v_\theta(\mathbf{z}_t, t)dt$.

Thus, our neural map $f$ involves changes of dimensionality, and evaluating its change of density when mapping samples requires a generalization of the standard change of variables formula in Equation 5. A simple way to see this is that the Jacobian for what we will proceed to term a *trans-dimensional mapping* is a rectangular matrix (not square) and hence does not have a well-defined determinant.

In obtaining a change of variables formulas for trans-dimensional mappings such as $E : X \mapsto \mathcal{Z}$ and $D : \mathcal{Z} \mapsto \mathcal{X}$, we consider an autoencoder's *decoder manifold*

$$\mathcal{M} = \{D(z) : z \in \mathcal{Z}\} \tag{12}$$

for which the change of variables formula will hold. Since we overfit our autoencoder on samples and do not require generalization to new data points, the points for which we evaluate the change of variables will lie in this manifold (assuming the size of the latent space and the expressivity of $E$ and $D$ are sufficient for encoding our dataset). If $E$ and $D$ are each other's inverse, then, for points on the decoder manifold $x_m \in \mathcal{M}$, the decoder's change of density between their projection $z = E(x_m)$ and their projection's *fibers* $\mathcal{F}(z) := \{x \in \mathcal{X} : z = E(x)\}$ is (Köthe, 2023)

$$\rho_{\mathcal{Z}}(z) = \rho_{\mathcal{X}}(\mathcal{F}(z))\sqrt{\left|\det(J_D^T J_D)\right|} \tag{13}$$

where $J_D$ is the decoder's Jacobian. For the encoder, for points on the decoder manifold and with $J_E$ as the encoder's Jacobian, the change of density is

$$\rho_{\mathcal{X}}(x) = \rho_{\mathcal{Z}}(z)\sqrt{\left|\det(J_E J_E^\top)\right|}. \tag{14}$$

These generalized change of variables formulae rescale a density by the mapping's Jacobian (or transposed Jacobian) volume (Ben-Israel, 1999): $\text{vol}J = \sqrt{\det J^T J}$. For square Jacobians of maps between same-dimensional spaces, $\text{vol}J = \sqrt{\det J^T J} = |\det J|$ which recovers the scaling factor of the standard change of variables formula (Equation 5).

### 3.3 FreeFlow Free Energy Difference Estimation

To obtain the change of variables between the two systems, we need to apply Equation 14 twice, once for mapping $\mathcal{X}_A$ to $\mathcal{Z}_A$ with the encoder $E_A$ and once for mapping $\mathcal{Z}_B$ to $\mathcal{X}_B$ with the decoder $D_B$. We also integrate the instantenous change of variables (Equation 8) over the latent continuous normalizing flow for our architecture to calculate the generalized energy difference as:

$$\Psi_F(\mathbf{x}) = U_B\left(D_B\left(f_z\left(E_A(\mathbf{x})\right)\right)\right) - U_A(\mathbf{x})$$

$$- \beta^{-1}\left(\underbrace{\log\left|\det\left(\mathbf{J}_{E_A}\mathbf{J}_{E_A}^\top\right)\right|^{-\frac{1}{2}}}_{\text{CoV } \mathcal{X}_A \to \mathcal{Z}_A} + \underbrace{\int_0^1 \text{tr}\left(\frac{\partial v(\mathbf{z}(t), t)}{\partial \mathbf{z}(t)}\right) dt}_{\text{CoV } \mathcal{Z}_A \to \mathcal{Z}_B} + \underbrace{\log\left|\det\left(\mathbf{J}_{D_B}^\top\mathbf{J}_{D_B}\right)\right|^{-\frac{1}{2}}}_{\text{CoV } \mathcal{Z}_B \to \mathcal{X}_B}\right)$$

$$\tag{15}$$

To estimate the free energy difference between systems A and B, we proceed as follows for training and inference:

**Training**

1. Run MD simulations for systems A and B to obtain sets of samples $\mathcal{X}_A$ from $\rho_A$ and $\mathcal{X}_B$ from $\rho_B$.

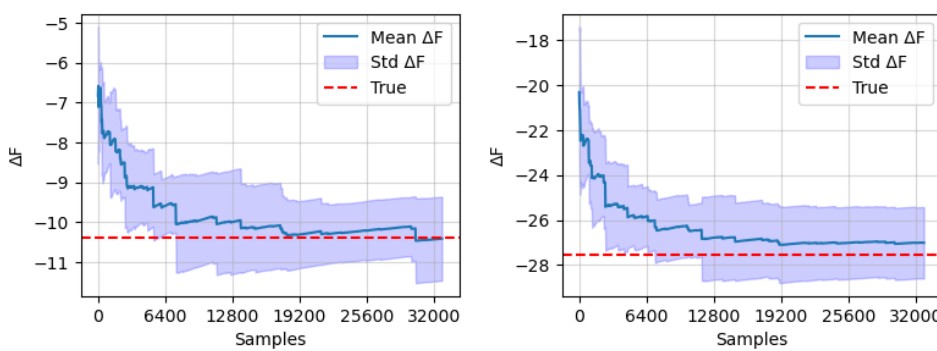

(a) Equidimensional Gaussian $\Delta F$ Estimates    (b) Transdimensional Gaussian $\Delta F$ Estimates

Figure 4: **Convergence of the $\Delta F$ estimates** between equidimensional and transdimensional Gaussians. The solid lines and the shaded regions show the mean and standard deviation of the estimates averaged over five runs.

2. Train the autoencoders $(E_A \circ D_A)$ on $\mathcal{X}_A$ and $(E_B \circ D_B)$ on $\mathcal{X}_B$.

3. Encode both sets of samples into the latent space to obtain $\mathcal{Z}_A$ and $\mathcal{Z}_B$.

4. Train the flow model using flow matching between $\mathcal{Z}_A$ and $\mathcal{Z}_B$, minimizing the objective in Equation 7.

**Inference**

1. Encode, integrate through the flow, and decode $\mathcal{X}_A$ to obtain approximate samples $\tilde{\mathcal{X}}_B$ from $\rho_B$. Compute $\Psi(\mathbf{x}_A)$ for $\mathbf{x}_A \in \mathcal{X}_A$

2. Use the $\Psi$ values to estimate $\Delta F$ with the TFEP estimator $\mathbb{E}_A\left[\exp(-\beta\Psi(x)\right] = \exp(-\beta\Delta F)$.

## 4 EXPERIMENTS

We evaluate FreeFlow on tasks of increasing complexity, starting with bridging Gaussian distributions of different dimensions, then two metastable states of the small molecule alanine dipeptide, and finally we bridge the Boltzmann distributions of different pairs of pharmaceutically relevant ligands with varying numbers of atoms.

### 4.1 GAUSSIAN DISTRIBUTIONS OF DIFFERENT DIMENSIONS

We first demonstrate that our generalized change of variables framework can be applied to transdimensional mappings, such as FreeFlow's encoder and decoder. Additionally, we aim to demonstrate the ability to bridge distributions with differing dimensionality. To address these two questions, we construct simplified toy problems using Gaussian distributions, which can be easily compressed to lower-dimensional spaces. These distributions allow us to compute the free energy difference analytically for reference values. Specifically, for two Gaussians of arbitrary dimensionalities with covariance matrices $\boldsymbol{\Sigma}_1, \boldsymbol{\Sigma}_2$, their free energy difference is the logarithmic ratio of their partition functions:

$$\Delta F = \log \frac{Z_2}{Z_1} = \log \frac{\sqrt{(2\pi)^{d_2}\det(\boldsymbol{\Sigma}_2)}}{\sqrt{(2\pi)^{d_1}\det(\boldsymbol{\Sigma}_1)}}$$
$$= \frac{1}{2}\left((d_2 - d_1)\log(2\pi) + \log\left(\frac{\det(\boldsymbol{\Sigma}_2)}{\det(\boldsymbol{\Sigma}_1)}\right)\right). \tag{16}$$

First, to validate the change of variables formulation, we let both system A and system $B$ be 30-dimensional zero-mean Gaussians with covariance matrices $\boldsymbol{\Sigma}_A = I$ and $\boldsymbol{\Sigma}_B = 0.5I$. Then, to

evaluate the trans-dimensional mapping, we let system A and B be zero-mean Gaussians with identity covariance, but in 60 and 30 dimensions. For both tasks, we use a latent space of 16 dimensions, and after sampling 50,000 samples from each distribution, we train the autoencoders for 100 and the flow model for 200 epochs.

Figure 4 shows the histogram of the energy distributions and the convergence of the $\Delta F$ estimates using FreeFlow. The convergence of the estimator towards the ground truth values empirically validates the use of the generalized energy difference of Equation 15 to bridge distributions of different dimensions. Thus by the convergence of the estimate in Figure 4a, we first empirically validate our modification of the generalized energy difference in Equation 15 between, and then by the convergence in Figure 4b, we conclude that the formulation also holds for trans-dimensional mappings. The estimate in Figure 4b exhibits a slight deviation in its mean from the true value. We believe to be due to the transdimensional change of variables formula being an approximation. More specifically, unless we can obtain a zero-loss autoencoder, there will be data points outside its decoder manifold and the change of variables formula will not hold exactly for those values.

## 4.2 METASTABLE STATES OF ALANINE DIPEPTIDE

After empirically validating our approach on toy cases, we evaluate if FreeFlow can be applied to a small physical system simpler than the larger molecules used in drug discovery tasks. For this purpose, we estimate the free energy difference between two metastable states of the small molecule alanine dipeptide. It is a small (32 atoms) yet non-trivial molecule commonly used as a benchmark in computational chemistry due to its well-known conformational dynamics. We distinguish the two metastable states with respect to the dihedral angle $\phi$, with system A $\phi \in [-\pi, 0] \cup [2.15, \pi]$ and system B to $\phi \in (0, 2.15)$.

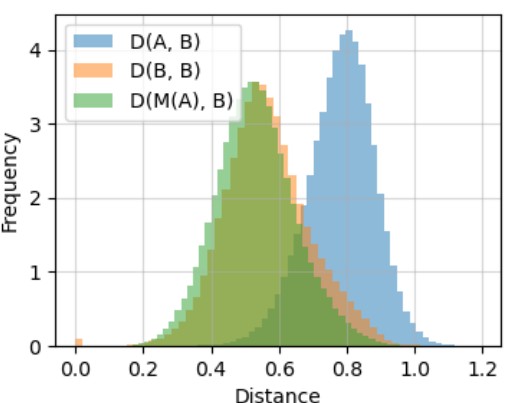

Using the OpenMM library (Eastman et al., 2017), we simulate alanine dipeptide in vacuum for 400 ns with step size 2 fs and save the state every 500 steps to obtain 400,000 samples in total, and then separate the source and target distributions with respect to the angle $\phi$. In the end, we obtain 371,094 source and 28,906 target samples. For the reference free energy difference, we use the values in (Invernizzi et al., 2022) obtained via OPES simu-

Figure 5: **Pairwise distances for alanine dipeptide samples.** $D(\cdot, \cdot)$ denotes the distribution of pairwise distances between two sets with $A, B$, the source and target systems, and $M(A)$, the set $A$ is mapped to via FreeFlow.

lations (Invernizzi, 2021) and estimations of the ratio of the partition functions of the two states.

Figure 5 displays the distributions of pairwise distances between samples from $A$, $B$, and the estimated samples $M(A)$, where for two sets $A$ and $B$, we define $D(A, B) := \{d(a, b) : a \in A, b \in B\}$ with $d$ being the Euclidean distance. We observe a strong agreement between the pairwise distances of samples from $B$ among themselves, and the distances between $B$ and the mapped samples $M(A)$, which is a desirable property for a flow model but not by itself sufficient to determine its accuracy. Similar to Figure 4b, the estimated pairwise distributions show a deviation from the true values, which we again attribute to the approximate nature of the trans-dimensional change of variables formula. Nevertheless the accuracy of the flow model is further supported by the estimate we obtain of $19.03 \pm 1.69$ kj/mol (averaged over five runs, $\pm$ standard deviation) compared to the reference of 20.87 kj/mol. We thus conclude that FreeFlow can be applied to physical systems before we move on to more relevant real-world use cases.

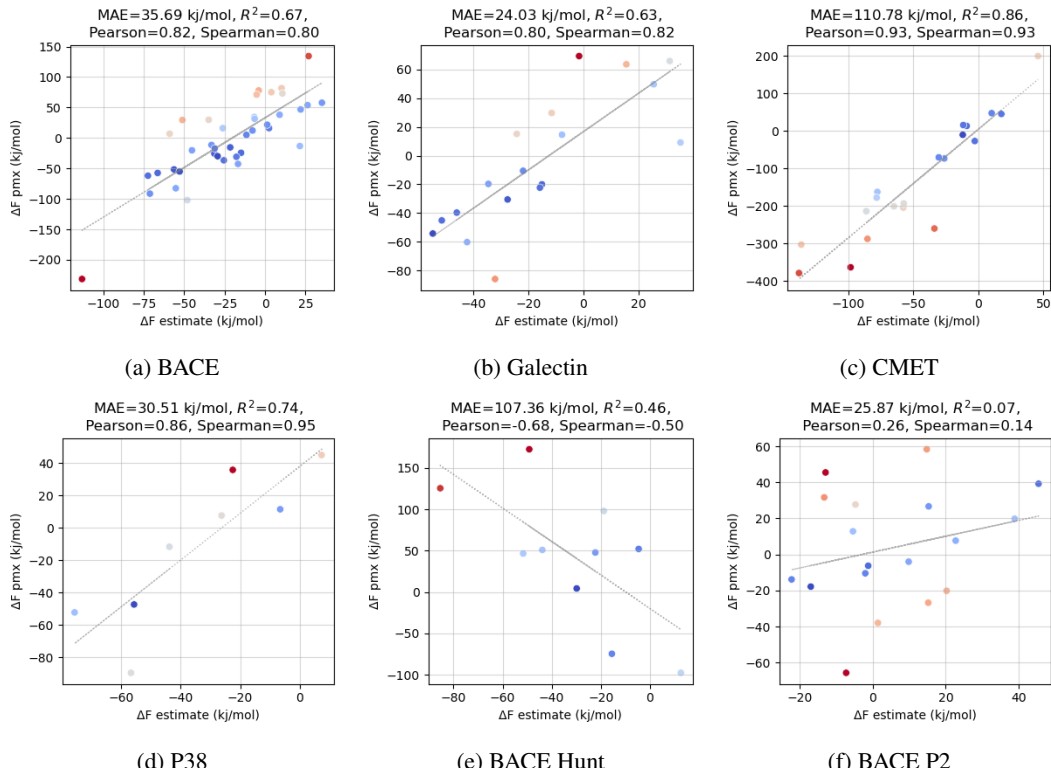

Figure 6: **Estimated and reference** $\Delta F$ **values (kj/mol) between ligands in water**, separated based on the subset of the Protein-Ligand Benchmark they belong to. Each dot represents one pair, with the x-axis denoting our estimates and the y-axis to the values calculated in the `pmx` library (Gapsys et al., 2020). The color of each dot corresponds to the absolute difference between its coordinates (red: higher, blue: lower), and the gray line is a linear regression fit to the points. We report various correlation measures as well as the mean absolute difference (MAE) above each plot.

### 4.3 PHARMACEUTICALLY RELEVANT LIGANDS IN SOLVENT

Finally, we evaluate FreeFlow on a real-world use case commonly addressed by FEP: comparing the binding free energies of ligands to a protein, a key task in drug discovery. This process involves two main legs: the solvent leg, where the free energy difference between the two ligands is estimated in solution, and the complex leg, where they are bound to the protein. For this evaluation, we focus on the solvent leg, using a set of pharmaceutically relevant ligands of varying sizes. More specifically for this task, the solvent leg involves system A, where one ligand is in water, and system B, where a different ligand is in water. The task is to learn a mapping between the Boltzmann distributions of these two systems in order to estimate the free energy difference between them.

**Data Collection and Reference Values.** We separately simulate each ligand in water at temperature 300 K for 400 ns with a step size of 2 fs using the OpenMM library (Eastman et al., 2017). As force field, we use the implicit `GBn2` solvation model (Nguyen et al.) with the `gaff-2.11` force field (Wang et al., 2004). We save a sample every 200 steps for a total of 1,000,000 samples from each ligand. The reference values we use are free energy differences from the `pmx` library (Gapsys et al., 2020) which were calculated using alchemical FEP (see Figure 2) and an explicit solvent potential. We use the OpenMM bridge within the `bgmol`[1] library to evaluate the energy functions. Before we train FreeFlow on a pair of ligands, we align each sample of the two ligands to a single reference by rotating and translating to minimize the root-mean-square distance between the sample and the reference. This minimizes the distance between the samples while leaving their potential energies

---

[1]https://github.com/noegroup/bgmol

unchanged, and makes training easier. We then train the two autoencoders for 500 epochs each, and the flow model for 200 epochs.

Figure 6 displays the agreement between the estimates we obtain and the reference values along with the resulting $R^2$ values, Pearson and Spearman correlations, and the mean absolute error between the estimates and the reference values. We acknowledge the high absolute error of the method, however, this can be attributed to some of the simplifications we made such as using an implicit solvent potential. Nonetheless, FreeFlow shows a very strong agreement for four of the six subsets of ligands with correlation coefficient greater than or equal to 0.8, which demonstrates its effectiveness in obtaining free energy differences between arbitrary ligands. These results indicate that FreeFlow can be beneficial in comparing relative binding free energies, an important real-world use case in drug discovery, where good correlation to reference values is necessary for accurate comparisons.

## 5   Conclusion

In this paper, we proposed FreeFlow, a novel method for estimating free energy differences between two systems by first encoding both systems into lower dimensional latent space, and training a flow model via Flow Matching to bridge the two latent distributions. This leads to fast training through the simulation-free regression objective of Flow Matching, and has the main benefit that free energy differences between systems of different dimensions can be estimated without resorting to nonphysical modifications such as dummy atoms. The trans-dimensional latent map not being invertible makes the typical formulations of change of variables inapplicable, and we build on previous work to solve this challenge by separating the change of variables among the three components of the map.

We evaluated FreeFlow first between simple Gaussian distributions to empirically validate our approach to learning a trans-dimensional map and our change of variables formulation. We then estimated the free energy difference between two states of the small molecule alanine dipeptide, which confirmed FreeFlow's applicability to physical systems. We finally estimated the free energy differences between pairs of pharmaceutically relevant ligands of various dimensionality in water, which represents one leg of the thermodynamic cycle commonly used to compare different molecules' binding affinities to a protein, a critical task in drug discovery.

We anticipate that as future work FreeFlow can be extended to the other leg of the thermodynamic cycle, learning a mapping between two bound protein-ligand complexes. This considerably increases the dimensionality of the problem but can be tackled by FreeFlow since the lower dimensional latent flow would still be fast to train.

## 6   Reproducibility Statement

We have implemented our method and experiments using publicly available libraries, primarily the PyTorch library (Paszke et al., 2019) for architectures and training, the `torchcfm` package (Tong et al., 2023) for an implementation of flow matching, and OpenMM (Eastman et al., 2017) for molecular dynamics simulations. All reference values were either computed by us as explained in the paper (e.g. in Section 4.1) or taken from the referenced resources. To reproduce the experiments, we report how the data was collected for each experiment in their respective subsections in Section 4, and we provide additional architectural and training setup details in Appendix A. We finally provide the steps required to use our method, as well as the trans-dimensional change of variables formulation in Section 3.3. Our full implementation of the method and the experiments will also be made available with the paper being publicly available.

## 7   Ethics Statement

We propose a method to estimate free energy differences between molecular systems, which is an important problem in drug discovery, particularly to compare the binding affinities of different ligands to proteins such as when screening a large number of candidate molecules to identify potential binders. The use of our method is not inherently ethical or unethical since it can be applied for a variety of goals, depending on the properties of the molecules and proteins involved. Nevertheless,

methods to accelerate the drug discovery process have immense potential benefits, especially when speed is a concern such as during a pandemic.

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

## A    EXPERIMENTAL DETAILS

### A.1    MODEL ARCHITECTURES

For the autoencoders, we implement both the encoders and the decoders as MLPs with four fully-connected layers with Scaled Exponential Linear Unit (SELU) activations (Klambauer et al., 2017), except for the final layer, which is linear to allow unbounded output values. Each hidden layer contains 128 neurons.

We construct the flow model as an MLP as well. It takes as input the flattened latent coordinates and the scalar time variable $t$, resulting in an input dimension of $d_{\text{latent}} + 1$. The flow model MLP also consists of four hidden layers, each with 64 units, and uses the Scaled Exponential Linear Unit (SELU) activation function (Klambauer et al., 2017) to promote self-normalizing properties in the network.

We use the Adam optimizer (Kingma & Ba, 2017) with a learning rate of $10^{-3}$ for all models, and set the batch size to 512 for all training runs as well as the mini-batch OT couplings within flow matching to simplify the implementation.

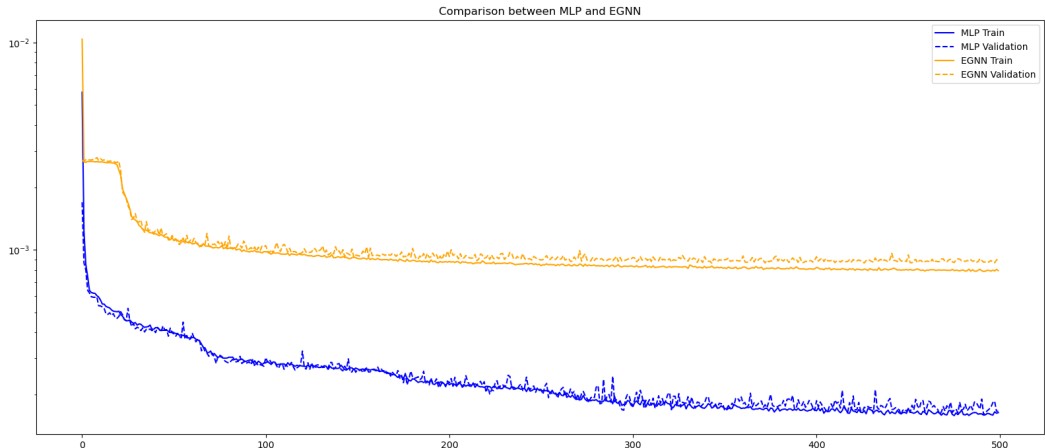

Figure 7: A comparison between the performance of the auto-encoder with different encoder architectures. The two models have roughly the same number of parameters and were trained for 500 epochs on alanine dipeptide conformations. It can be clearly seen that the MLP outperforms the EGNN by almost an order of magnitude in terms of reconstruction error. Furthermore, the MLP-based encoder is eight times faster than the EGNN-based one, which is extremely relevant for our method.

## A.2 FLOW MATCHING SETUP

We utilize OT couplings, approximated via mini-batches as proposed by Fatras et al. (2021), to construct the coupling between samples from the source and target latent distributions. OT couplings are advantageous because they lead to straighter transport paths, which can be integrated more efficiently with lower numerical integration error (Tong et al., 2023; Klein et al., 2023). Additionally, the use of OT couplings reduces the variance of the CFM objective since samples $x_0 \sim \rho_0$ are more likely to be coupled with nearby samples $x_1 \sim \rho_1$, rather than with samples drawn uniformly from $\rho_1$. We then take the linear vector field $u_t = x_1 - x_0$ as the regression target, and use Gaussian probability paths with $\rho_t(x) = \mathcal{N}(x; (1-t)x_0 + tx_1, \sigma^2)$ where we set $\sigma = 10^{-4}$.

## B DERIVATION OF THE TARGETED FEP ESTIMATOR

As proposed in (Jarzynski, 2002), free energy differences can be estimated by mapping the source distribution $A$ to an approximation $B'$ of the target distribution $B$ via the mapping $M$ and doing importance sampling from $B'$ to $B$. We now show that the equality in Equation 3 holds:

$$\mathbb{E}_A \left[ e^{-\beta \Phi_F} \right] = \int_A \rho_A(x) e^{-\beta \Phi_F(x)} dx \tag{17}$$

$$= \frac{1}{Z_A} \int_A e^{-\beta U_A(x) - \beta \Phi_F(x)} dx \quad \text{since} \quad \rho_A(x) = \frac{e^{-\beta U_A(x)}}{Z_A} \tag{18}$$

$$= \frac{1}{Z_A} \int_A e^{-\beta U_A(x) - \beta U_B(M(x)) + \beta U_A(x) + \log |J_M(x)|} dx \tag{19}$$

$$= \frac{1}{Z_A} \int_A e^{-\beta U_B(M(x))} |J_M(x)| dx \tag{20}$$

$$= \frac{1}{Z_A} \int_B e^{-\beta U_B(y)} dy \quad \text{after change-of-variables with } y = M(x) \tag{21}$$

$$= \frac{Z_B}{Z_A} \tag{22}$$

$$= e^{-\beta \Delta F} \quad \text{since} \quad \Delta F = -\log \frac{Z_B}{Z_A}. \tag{23}$$

## C  DERIVATION OF FREE ENERGY AS LOGARITHM OF PARTITION FUNCTION

Given the probability distribution $\rho(x) = \frac{-\beta U(x)}{Z}$ with energy function $U(x)$ and partition function $Z = \int_x e^{-\beta U(x)}$ where $\beta = \frac{1}{kT}$ is the inverse temperature, we have the internal energy $U$ of the system

$$U = \int_x \rho(x) U(x) = \int_x \frac{e^{-\beta U(x)}}{Z} U(x) \tag{24}$$

and entropy $S$ defined as

$$S = -k \int_x \rho(x) \ln(\rho(x)) = -k \int_x \frac{e^{-\beta U(x)}}{Z} \ln\left(\frac{e^{-\beta U(x)}}{Z}\right). \tag{25}$$

By algebraic manipulations and using the definitions above, we can obtain

$$S = k\beta U + k \ln(Z). \tag{26}$$

The Helmholtz free energy $(F)$ of a system is defined as

$$F = U - TS = U - T(k\beta U + k \ln(Z)). \tag{27}$$

If we then plug in the definitions above and simplify using $\beta = \frac{1}{kT}$, we obtain

$$F = -kT \ln(Z) \tag{28}$$

which concludes the derivation of free energy as the logarithm of the partition function.

