# OpenReview forum: "FreeFlow: Latent Flow Matching for Free Energy Difference Estimation"
_ICLR.cc/2025/Conference — Submitted to ICLR 2025_

### Official Review · Reviewer_yw5g · 2024-11-01

**Soundness:** 3
**Presentation:** 3
**Contribution:** 3
**Rating:** 8
**Confidence:** 4

**Summary:**

The authors propose a strategy to estimate free energy differences between molecular systems. Their approach relies on performing latent diffusion (flow matching) instead of diffusion in data space. The main advantage is that, due to the change in dimensionality required to match different molecular systems, diffusion in data space requires the addition of dummy atoms. In contrast, diffusion in latent space is dimension-agnostic. The trade-off is the need to track all the change-of-variable factors that come with the steps to match the two distributions: encoding, diffusion, and decoding.

The authors provide experiments to demonstrate the validity of their approach, using both toy distributions of multi-dimensional Gaussians and more real-world applications.

**Strengths:**

The paper is well-written with clear and consistent notation, and the topic is of relevance in the field of drug discovery. The authors effectively explain existing methods while clearly stating and motivating their contribution.

**Weaknesses:**

The paper's main weakness is that it doesn't develop an entirely new method, but rather combines existing methodologies into a framework for free energy difference estimation. In the conclusions, the authors mention plans to apply this methodology to learn a mapping between bound protein-ligand complexes in future work. From a drug discovery perspective, this would be a more significant goal. I wonder if it's feasible for the authors to present the full thermodynamic cycle in this work.

Additionally, while instructive, the experiments are based on a limited set of examples. A broader set of experiments to evaluate their method on a large-scale dataset would definitely improve the paper.

**Questions:**

- I understand that overfitting the autoencoder for density reconstruction is auxiliary to performing flow matching in latent space. However, I wonder if there could be benefits to learning a "representation learning" autoencoder in the classical sense—one that can generalize and thus be used for all densities rather than requiring separate autoencoders for each molecular system.
- In Figure 4(a), the true target and the estimated target distributions differ significantly. In fact, the true target seems to coincide with the source. Could this be a labeling mistake?
- I'm not entirely clear on the theory (or implications) of the trans-dimensional change of variable. Let's consider a simple case, as depicted in Figure 3 (bottom). We have an encoder (in the paper's terminology) $f:\mathbf{R}^2 \rightarrow \mathbf{R}$, and we'll omit the decoder, leaving only this mapping.
    - The Jacobian is $J_f(x) = (\frac{\partial f}{\partial x_1}, \frac{\partial f}{\partial x_2})$, so the volume form is simply the square root (the determinant being irrelevant since this is one-dimensional) of $\sum_i \left(\frac{\partial f}{\partial x_i}\right)^2$, correct?
    - Given these assumptions, is the equation in Figure 3 valid for all x? This seems counterintuitive, as for generic $\rho_A$ and $f$, the mapping won't be lossless. For instance, the equation in the top part of Figure 3 holds if $f$ is a bijection.
    - If that's not the case, what's the most general statement we can make for this scenario (namely, $f:\mathbf{R}^2 \rightarrow \mathbf{R}$ with no decoding)?
    - For a surjective $f$, does the reverse equation hold true? That is, $\rho_B(f(x)) = \rho_A(x) |\det J^T J|^{1/2}$? This seems more plausible, as even if $f$ isn't lossless, I should be able to make a statement about $\rho_B(f(x))$ knowing $f$ and $\rho_A$.

---

> ### Author Response · Authors · 2024-11-23
>
> Thank you for the detailed review!
>
> ---
> **“The paper [...] doesn't develop an entirely new method, but rather combines existing methodologies into a framework for free energy difference estimation.”**
>
> The technical novelty of our work lies not in using a flow to bridge the distributions, but rather in using a latent flow that employs a trans-dimensional change of variables formula to account for dimensionality changes between systems with different numbers of atoms, eliminating the need for dummy atoms and enabling efficient free energy difference estimation in molecular systems. We see our work as an initial and early, yet important step towards enabling free energy estimations through a latent space framework.
>
> **“I wonder if there could be benefits to learning a "representation learning" autoencoder in the classical sense—one that can generalize and thus be used for all densities rather than requiring separate autoencoders for each molecular system.”**
>
> Thanks for sharing this idea. Exploring a generalizable autoencoder could be an exciting direction for future research. An autoencoder accurate across a set of molecules would save the time to train separate autoencoders and therefore speed up the free energy difference calculations. Nevertheless, this could be challenging as such an autoencoder would need to learn a more complex mapping, requiring a large-scale molecular dataset and costly training. Our current approach focuses on overfitting to maximize accuracy and efficiency for specific pairs of molecules.
>
> **“In Figure 4(a), the true target and the estimated target distributions differ significantly. In fact, the true target seems to coincide with the source. Could this be a labeling mistake?”**
>
> Thanks for pointing this out. Upon closer examination, we found that this was not a labeling mistake. The observed phenomenon arises because points sampled from different same-dimensional isotropic Gaussian distributions inherently exhibit the same energy distribution when evaluated using their respective energy functions. As a result, our plots were inappropriate to measure the quality of the map. To address this issue, we decided to remove this plot.
>
> **Theory (or implications) of the trans-dimensional change of variable**
>
> It is indeed true that the map is not lossless, and the change of variables formula is an approximation (for similar work see [1]). This approximation error is minimized to the extent that our data manifold (points we compute the change of variables for) overlaps with the decoder manifold (image of the decoder). We try to minimize this approximation error by overfitting the autoencoders to the data and our results show that this is a promising approach to evaluating free energy differences.
>
> ---
> [1] Gemici, M. C., Rezende, D., & Mohamed, S. (2016). Normalizing Flows on Riemannian Manifolds (No. arXiv:1611.02304). arXiv. https://doi.org/10.48550/arXiv.1611.02304

---

> > ### Comment · Reviewer_yw5g · 2024-11-26
> >
> > I thank the authors for addressing some of my concerns. However, I asked some more detailed questions in the "Questions" section that were only very approximately touched upon. I look forward to a more detailed discussion.

---

> ### Author Response · Authors · 2024-12-01
>
> Thank you for following up on the discussion. We are happy to provide further clarifications for the questions raised.
>
> **"The Jacobian is $J_f(x) = (\frac{\partial f}{\partial x_1}, \frac{\partial f}{\partial x_2})$, so the volume form is simply the square root (the determinant being irrelevant since this is one-dimensional) of $\sum_i \left( \frac{\partial f}{\partial x_i} \right)^2$, correct?"**
>
> Yes. There is no Jacobian determinant and the square root of the gradient vector is not the volume change. The change in density can again be computed using Equation 14.
>
> **"Given these assumptions, is the equation in Figure 3 valid for all x? This seems counterintuitive, as for generic
>  and $\rho_A$ and $f$, the mapping won't be lossless. For instance, the equation in the top part of Figure 3 holds if
>  is a bijection.
> "**
>
> Yes, the equation in Figure 3 (top) only holds if $f$ is a bijection, and we had intended to show this in the figure.
>
> Since the encoder cannot be a bijection and will be lossy, the change of density is also an approximation. Intuitively, an encoding $z$ collects the probability mass of all $x$ such that $f(x) = z$. This means that we can obtain a meaningful density function over the latent space just using the encoder.
>
> However, if we want to obtain an ambient density from a latent density, the best we can get is a density defined over the fibers $\mathcal{F}(z)$ of the decoder (Equation 13, note that input to $p_X$ is $\mathcal{F}(z)$), where $\mathcal{F}(z) := \{x : f(x) = z\}$.
>
> The decoder change of variables is then an approximation as we are not able to distinguish the densities of the points within the same fiber. This is not an issue if the decoder manifold (decoder’s image) overlaps perfectly with the data manifold, i.e. the autoencoder reconstructs the data samples perfectly, because then we could have a density function defined for each point we will be evaluating it on.
>
> Of course this is hard to achieve in practice. There will then be data points outside the decoder manifold, and the density function we get will not be able to distinguish between them.
>
> **"For a surjective $f$, does the reverse equation hold true?"**
>
> In the case of a surjective mapping, the fibers $F(z) = \{x : f(x) = z\}$ will have multiple elements. In this scenario, the density $p_B(f(x))$ in the latent space accounts for the aggregated probability mass of all points $x$ that map to the same $z$. The equation will be exact even though $f$ is lossy, since each latent $z$ will aggregate the probabality mass in $F(z)$.

---

### Official Review · Reviewer_pMxg · 2024-11-03

**Soundness:** 2
**Presentation:** 3
**Contribution:** 2
**Rating:** 3
**Confidence:** 4

**Summary:**

In this paper, the authors focus on estimating the free energy difference between two molecular systems. They propose to use a neural network to learn a mapping between these two systems, which can reduce the variance of free energy estimation based on TFEP, providing more accurate result compared to FEP. The experiments on several simple cases demonstrate the effectiveness of proposed method. On the large molecules, the authors also claim that the result from proposed method is close to the reference data.

**Strengths:**

1. Using neural networks to represent the mapping between different distributions may bring more flexibility to the solution.
2. The authors give a lot of examples that help to understand the proposed method.
3. The writing is good.

**Weaknesses:**

1. The accuracy of the proposed method is not good enough. In the experiments of large molecules, the MAE between the proposed method and the baseline method is about tens of or even hundreds of kJ/mol. As a comparison, the error of free energy should be within 10kcal/mol (~42kJ/mol) to give a qualitatively correct prediction. Such a large difference means that the proposed method may not be reliable in practice. The authors should consider how to improve the accuracy of the proposed method.
2. The cost of the proposed method is relatively large. The training data includes molecular dynamics simulation of the related systems. Thus, when applying the proposed method to new systems, one should perform additional molecule dynamics simulation to collect training data. Given the poor accuracy of the proposed method, the reviewer believes the training cost is larger than expected.

**Questions:**

Suggestions are listed in weakness.

---

> ### Author Response · Authors · 2024-11-23
>
> Thank you for your feedback!
>
> ---
> **Accuracy of the proposed method**
>
> While the absolute error of our method is high, we would like to highlight that our work is an early attempt at estimating free energy differences using ML approaches, to alleviate the shortcomings of the classical methods. Our main contribution is that using a latent flow to address the problem of systems with different dimensionalities is a promising direction as an alternative to using dummy atoms. There is certainly more work needed to make the approach applicable and useful in practice.
>
> Additionally, we model our systems using an implicit solvent rather than modeling the solvent atoms explicitly as that would lead to systems with an extremely high dimensionality and thus a harder learning problem. Using an implicit solvent introduces an additional source of error, and that is one the reasons why we focus on correlations rather than absolute errors in our evaluation. Correlations are still useful as free energy differences are often used to compare binding affinities which rely on the relative values between two candidates.
>
> **“The cost of the proposed method is relatively large. The training data includes molecular dynamics simulation of the related systems. Thus, when applying the proposed method to new systems, one should perform additional molecule dynamics simulation to collect training data.”**
>
> We have clarified in our Introduction that FreeFlow indeed still requires simulation but the amount required is an order of magnitude lower than for alchemical FEP for which one needs to perform simulations for multiple intermediate states connecting the two states.

---

### Official Review · Reviewer_ZU4w · 2024-11-04

**Soundness:** 2
**Presentation:** 3
**Contribution:** 3
**Rating:** 6
**Confidence:** 3

**Summary:**

The authors present FreeFlow, a novel method for learning normalizing flows between distributions of arbitrary dimensions. FreeFlow achieves this using flow matching to learn an invertible map between distributions embedded into a lower dimensional latent space. The authors demonstrate the use-case of FreeFlow in application to estimating free energy differences for molecular systems. Through empirical experiments, FreeFlow is shown to efficiently estimate free energy differences for molecular systems.

**Strengths:**

This work addresses the challenging problem of estimating free energy differences between molecular systems. In doing so, the authors devise a framework for learning normalizing flows between distributions of arbitrary dimensions. The introduced method (FreeFlow) has two key strengths:
- FreeFlow leverages advancements in flow matching to learn more expressive maps between molecular distributions.
- FreeFlow does not require the use of non-physical modifications on molecules to match dimensions between distributions.

**Weaknesses:**

A central limitation of this work is that there appears to be a lack of comparisons with existing approaches that estimate free energy differences between molecular systems. Without comparison to other methods, it is hard to assess the validity of some of the claims and contributions of this work. For example:
- The authors argue that FreeFlow learns more expressive maps between molecular distributions compared to previous normalizing flows solutions. It is not clear that this claim is supported in the experiments. What are the previous normalizing flows solutions that FreeFlow is compared against?
- One argued advantage of FreeFlow is that the method does not require the use of non-physical modifications, such as the use of dummy variables on molecules to match the dimensions between distributions.  It is not clear how or if avoiding the use of non-physical modifications leads to improved free energy difference estimation. How does FreeFlow compare to methods that use such non-physical modifications? Likewise, why not compare to other baselines in that address the problem of free energy difference estimation (such as those described in Figure 2)?

In addition, there are several areas in the manuscript that contain some unaddressed items (see questions below) that at times make it difficult to follow the work. I believe that adding fair comparison to existing approaches for estimating free energy differences between molecular systems and addressing the questions I outlined below would strongly improve this work.

**Questions:**

- Lines 232-233: Could the authors provide some explanation/justification an optimal transport map between $\rho_A$ and $\rho_B$ is needed in this setting? What happens if you were to assume independent marginals?
- Regarding the latent representations:
    -  Do you validate that the auto-encoders are in fact over-fitting? Do you validate that the auto-encoders learn reasonable representations?
    - How did you decide on hyper-parameters for the auto-encoders?
    - The authors justify choice of MLP for the auto-encoder architecture as "not needing to generalize" (lines 257-259). How do you think the results would change if a different architecture is used, for examples a graph neural network or transformer, which have become de facto architectures for learning molecular representations?
- In Figure 4.a, is the objective to match the estimated target distribution (green) to the true target distribution (orange)? In which case, it appears this the estimate does not match the target. Is there intuition on why the estimated target distribution is so far from the true target distribution in the Equidimensional Gaussian Energy setting compared to the Transdimensional Gaussian Energy setting? I would assume that the Equidimensional setting should be easier?
- In Figure 5, what is the distribution pairwise distance $D(\cdot, \cdot)$?
- Lines 419-420: How do you observe the distances between $B$ and the mapped sampled $M(B)$? In Figure 5, only $D(A, B)$, $D(B, B)$, and $D(M(A), B)$ are reported.

---

> ### Author Response · Authors · 2024-11-23
> **Official Comment by Authors (Part 1/2)**
>
> Thank you for the review! We hope we are able to address the questions raised. Our response is in two parts due to the character limit:
>
> ---
> **“The authors argue that FreeFlow learns more expressive maps between molecular distributions compared to previous normalizing flows solutions. It is not clear that this claim is supported in the experiments.”**
>
> We have updated our explanation in the Introduction to reflect that by “expressive” we refer mainly to the less restricted design space enabled by flow matching compared to previous normalizing flows which depended on architectures with tractable Jacobians and fast invertibility. Flow matching and continuous normalizing flows on the other hand can be used with any typical neural network architecture.
>
> **“It is not clear how or if avoiding the use of non-physical modifications leads to improved free energy difference estimation. How does FreeFlow compare to methods that use such non-physical modifications?”**
>
> Free Energy Perturbation (FEP) which we compare with uses such “non-physical” alchemical intermediates and dummy atoms. A main benefit of our method is that it can be faster because it does not require the intermediate states and because it transforms the distributions in a lower dimensional space.
>
> **Comparisons to other baselines such as those described in Figure 2**
>
> We have compared FreeFlow with a normalizing flow consisting of coupling layers as in TFEP [2] on our alanine dipeptide task and observed significantly larger estimation errors with estimates on the order of 1,000 kj/mol compared to the reference value of 20.87 kj/mol, as well as unstable likelihood-based training due to the force-field-based potential energies.
>
> In our paper, we compare FreeFlow against alchemical FEP in our main results due to the practical relevance of the method and because there exist works that provide reference ΔF values for the solvent leg (between two unbound ligands) of the thermodynamic cycle.
>
> **“Lines 232-233: Could the authors provide some explanation/justification an optimal transport map between ρA and ρB is needed in this setting? What happens if you were to assume independent marginals?”**
>
> We thank you for your suggestion and now clarify in Section 3.1 that the primary benefit of approximating an OT map is that the vector field learned with mini-batch OT couplings will have straighter trajectories compared to one learned with independent couplings, and this will reduce the integration error. Specifically for free energy estimation, it has been shown using lower-dimensional systems in [1] that approximating an OT map between ρA and ρB leads to paths with lower free energy compared to linear interpolation, improving the convergence of the estimator.
>
> **Auto-Encoders: Validation of over-fitting and reasonable representations & decision on hyper-parameters**
>
> We have explained more clearly in Section 3.2 that we train separate autoencoders for each molecule until the mean-squared reconstruction loss on training data converges, since we will be using the training data to estimate ΔF. Additionally for the hyperparameters, we performed validation runs on alanine dipeptide and kept the best-performing ones.
>
> **“How do you think the results would change if a different architecture is used, for examples a graph neural network or transformer, which have become de facto architectures for learning molecular representations?”**
>
> Thank you for pointing out the significance of comparisons against graph neural networks. We have now added Figure 7 in Appendix A showing that an MLP achieves an almost an order of magnitude lower reconstruction loss than an EGNN with a similar number of parameters. Furthermore, the training time for the MLP is significantly lower (2 min.) compared to the EGNN (17 min.).
>
> We believe this to be because we can overfit on the samples and we have sufficient data, not requiring us to have a more complex geometry-aware model. The EGNN is a more complex function and there’s some optimization error introduced by using it. Furthermore, using a GNN significantly slows down the training procedure (over eight times slower than the MLP) which is detrimental to our approach since our main goal is to speed-up drug screening.

---

> ### Author Response · Authors · 2024-11-23
> **Official Comment by Authors (Part 2/2)**
>
> **Figure 4.a: Estimated target distribution is far from true target (equidimensional case)**
>
> We agree that the figures are not informative and removed them.
>
> Firstly, the first plot showed almost the same energy distribution for both source and target which is correct but not useful for our analysis. This is because points sampled from different same-dimensional isotropic Gaussian distributions exhibit the same energy distribution when evaluated using their respective energy functions. It would be better to plot the points from the source Gaussian under the target Gaussian’s energy function as we do for the mapped and target samples. However, this is not possible for the trans-dimensional case because the source points are of different dimensionality.
>
> We do not have a good explanation for why the energy of the mapped samples in the equi-dimensional case is matching the distribution of the target worse than in the trans-dimensional case. Because of the above and because we are mainly interested in the final free energy difference accuracy, we decided to remove these plots.
>
> **“In Figure 5, what is the distribution pairwise distance D(⋅,⋅)?”**
>
> We have made it clearer in the Section 4.2 that  by D(A,B) we denote the distribution of distances for all pairs (a,b) from A and B; i.e. the set {d(a, b) : a \in A, b \in B}.
>
> **“Lines 419-420: How do you observe the distances between B and the mapped sampled M(B) ? In Figure 5, only D(A,B), D(B,B), and D(M(A),B) are reported.”**
>
> The “M(B)” in lines 419-420 is indeed a typo and we have now fixed this. Since we map the samples from A with our flow, we only observe D(M(A), B) and not D(M(B), B).
>
> ---
> [1] Decherchi, S., and A. Cavalli. "Optimal Transport for Free Energy Estimation." The journal of physical chemistry letters 14.6 (2023): 1618-1625.
>
> [2] Wirnsberger, Peter, et al. "Targeted free energy estimation via learned mappings." The Journal of Chemical Physics 153.14 (2020).

---

> > ### Comment · Reviewer_ZU4w · 2024-11-25
> >
> > I thank the authors for their detailed responses to my comments, questions, and concerns. Thank you for clarifying my questions. In general, I am happy with the authors' rebuttal and the amendments made to the manuscript. With this, I am happy to raise my score.

---

### Official Review · Reviewer_rKcy · 2024-11-04

**Soundness:** 3
**Presentation:** 3
**Contribution:** 3
**Rating:** 5
**Confidence:** 5

**Summary:**

They propose a method for estimating free energy differences between molecular systems. Traditional free energy estimation methods rely on simulating intermediate states between two systems. FreeFlow addresses this by mapping both systems to a common latent space via autoencoders and applying a neural flow model to estimate free energy differences without intermediate simulations. This latent-space approach leverages flow matching to track density changes between systems, allowing FreeFlow to handle trans-dimensional mappings.

**Strengths:**

They propose a transdimensional mapping for free energy estimates. Super cool.
Their theory seems to directly relate to their implementation methods.

**Weaknesses:**

I would like to see way more comparisons with other methods!
Also in Figure 4, the mean of equidimensional gaussians seems to converge to the true mean, but not for the trans dimensional one.
In Figure 5, the D(B,B) and D(M(A),B) do not completely overlap.
Figure 6 is important, but more important is the comparison to existing methods. Free energy estimation is extremely difficult, so it is more interesting to see how much it improves upon other methods vs absolute accuracy.

**Questions:**

Why do the figures differ as mentioned in weaknesses?
Could you write out the proof of equation 13?
Also, please give at least a toy system code to test the method. Otherwise, it is difficult to interpret accurately.

---

> ### Author Response · Authors · 2024-11-23
>
> We thank the reviewer for the valuable comments and questions!
>
> ---
> **Toy system code to test the method**
>
> We added the link to our paper pointing to an example application of FreeFlow to Gaussian distributions  https://anonymous.4open.science/r/freeflow-example-26D4
>
> **Comparisons with other methods**
>
> The primary used methods for FED calculations involve the BAR estimate and Free Energy Perturbation (FEP) which are time-costly because they rely on intermediate transformations between the source and target state for accurate estimates. In our work we compare our estimates against values derived through the common FEP method. Flow-based FEP solutions like FreeFlow optimize the process of FED calculations by reducing the reliance on extensive intermediate transformations which we model through a CNF.
>
> We also trained a normalizing flow consisting of coupling layers as in TFEP [1] for our alanine dipeptide task and obtained significantly less accurate estimates for ΔF. The estimated values were on the order of 1,000 kj/mol compared to the reference value of 20.87 kj/mol. We also observed that the training was highly unstable due to the likelihood training using force-field-based potential energies.
>
> **“In Figure 4, the mean of equidimensional gaussians seems to converge to the true mean, but not for the trans dimensional one.”**
>
> We have added to the discussion in Section 4.1 that free energy difference estimates obtained from our trans-dimensional change of variables only hold to an approximation, since the encoders are lossy compressors. More specifically, unless we can obtain a zero-loss autoencoder that perfectly recovers the data points, there will be points outside its decoder’s image (the decoder manifold) and particularly for those points the change of variables formula will not hold exactly.
>
> **“In Figure 5, the D(B,B) and D(M(A),B) do not completely overlap”**
>
> We now clarify in Section 4.2 that the trans-dimensional change of variables formula being an approximation is likely also the reason behind this slight mismatch between D(M(A), B) and D(B, B). However, while we try to learn a map that accurately samples the target distribution, we (and TFEP more generally) can tolerate slightly inaccurate maps since the ultimate goal is to estimate ΔF via importance sampling. In fact, the importance sampling reweighting holds for any map (provided the energy functions are defined everywhere) and results in an unbiased estimator, but our goal is to learn a reasonably accurate map that leads to fast convergence of the ΔF estimate.
>
> ---
> [1] Wirnsberger, Peter, et al. "Targeted free energy estimation via learned mappings." The Journal of Chemical Physics 153.14 (2020).

---

### Author Response · Authors · 2024-11-23

We would like to thank the reviewers for their valuable feedback, which has helped us improve our paper. As we explain in more detail in the individual responses, we have made the following key revisions which are highlighted blue in the paper:

* Provided code for Gaussian example: added a link to our code to help readers test our method (https://anonymous.4open.science/r/freeflow-example-26D4)
* Clarified trans-dimensional change of variables: expanded explanations in Sections 4.1 and 4.2 about how autoencoder loss affects our estimates.
* Explained pairwise distance  D(*,*)): clarified in Section 4.2 that D(A,B) represents distances between samples from A and B.
* Justified optimal transport map: added reasoning in Section 3.1 for approximating an optimal transport map to improve our estimator.
Detailed autoencoder training: described our training process and compared MLPs with EGNNs, showing MLPs perform better (see Figure 7 in Appendix A).
* Removed uninformative plots: eliminated Figures 4(a) and 4(c).
* Addressed accuracy concerns: acknowledged limitations but emphasized our contribution as a promising initial approach.

---

### Meta-Review · Area_Chair_oGzp · 2024-12-21

**Metareview:**

This paper introduces FreeFlow, a method for estimating the free energy of a molecular system. This is a crucial problem, as free energy calculations are commonly performed in approximating / understanding protein-ligand binding strength and other molecular tasks. Different from common methods which require MD simulations online at test time, FreeFlow encodes the molecular systems in low dimensional continuous latent spaces and then uses flow matching in the latent space. The overall idea in the paper seems quite interesting, as it circumvents the need to add non-physical "dummy atoms" to one of the system in order to match dimensionality and, once trained, the model promises much better computational efficiency.

The primary thing missing from the paper seems to be experimental validation that shows the authors method estimates free energy in a reliable enough fashion that it is useful in some downstream task. Absent a large existing body of work using machine learning to estimate free energy, many of the results in the paper exist "in a vacuum" so to speak, where the authors' method is primarily evaluated through stand-alone MAE and correlation measures. As Reviewer pMxg points out, the absolute MAEs are relatively large. I fully agree with the authors point that rank correlation is in theory the more interesting metric here compared to absolute error, but rank correlation is also flawed in a vacuum -- it is possible to achieve relatively good rank correlation but still being able to resolve large absolute differences, the value of which is less clear in downstream tasks like protein-ligand binding prediction.

My proposal to resolve this reviewer concern in a resubmission would be to simply evaluate FreeFlow in a downstream task. Showing that FreeFlow's predictions are correlated with in silico oracles used in downstream tasks to solve problems like binding and that it could therefore be used as a proxy in these settings if we imagined the existing specialized ML oracles didn't exist would silence these concerns.

**Additional Comments On Reviewer Discussion:**

In response to reviewer questions, the authors made a number of substantial updates to their paper as helpfully detailed in their own summary comment. I think the remaining unaddressed concerns deal primarily with experimental validation, as I detail above.

---

### Decision · Program_Chairs · 2025-01-22

Reject